# Two New Phenolic Glycosides with Lactone Structural Units from Leaves of *Ardisia crenata* Sims with Antibacterial and Anti-Inflammatory Activities

**DOI:** 10.3390/molecules27154903

**Published:** 2022-07-31

**Authors:** Huihui Tao, Yongqiang Zhou, Xin Yin, Xin Wei, Ying Zhou

**Affiliations:** College of Pharmacy, Guizhou University of Traditional Chinese Medicine, Guiyang 550025, China; thlc922@163.com (H.T.); yinxin110901@163.com (X.Y.); sfweixin@163.com (X.W.)

**Keywords:** *Ardisia crenata* Sims, lactone, antibacterial, anti-inflammatory, natural product

## Abstract

Two new lactones, named Ardisicreolides A–B (**1**–**2**), together with four known flavonoids, Quercetin (**3**), Myricetrin (**4**), Quercitrin (**5**), Tamarixetin 3-O-rhamnoside (**6**), were isolated from the ethyl acetate portion of 70% ethanol extracts of dried leaves from *Ardisia crenata* Sims. These compounds were identified from *Ardisia crenata* Sims for the first time. The structures of **1**–**6** were elucidated according to 1D and 2D-NMR methods and together with the published literature. All of the isolated compounds were evaluated for in vitro anti-microbial effect against *Escherichia coli*, *Pseudomonas aeuroginosa*, *Enterococcus faecalis*, *Proteus vulgaris*, *Staphylococcus aureus*, and *Bacillus subtilis*. In addition, compounds **1**–**2** were assessed for anti-inflammatory activity by acting on LPS-induced RAW 264.7 macrophage cells in vitro. The results showed that only compound **2** exhibited moderate antibacterial activity on *Bacillus subtilis*. Moreover, compounds **1** and **2** were found to significantly inhibit the production of nitric oxide (NO) and reduce the release of tumor necrosis factor-α (TNF-α), interleukin-1β (IL-1β), interleukin-4 (IL-4), and interleukin-10 (IL-10) in LPS-induced RAW 264.7 macrophage cells. The present data suggest that lactones from the leaves of *A. crenata* Sims might be used as a potential source of natural anti-inflammatory agents.

## 1. Introduction

There are about 500 species of plants in the genus *Ardisia*, which are widely distributed in subtropical and tropical regions [1,2]. The *Ardisia crenata* Sims is a common evergreen shrub belonging to *Ardisia* of Myrsinaceae with red fruits at maturity [3]. In China, the roots of the red-fruited *A. crenata* Sims are used as a traditional Chinese medicine “Zhushagen” [4], which is widely used for the treatment of respiratory infections, toothache, arthralgia, menstrual problems, and fertility regulation [5,6,7]. These pharmacological activities are often closely related with different kinds of chemical constituents from the roots of *A. crenata* Sims. Previous phytochemical studies on the roots of *A. crenata* Sims have mainly revealed active constituents including triterpenoid saponins, coumarins, phytosterols, and benzoquinones [8,9,10]. Recently, researchers have extensively investigated these compounds for their anti-tumor, immunosuppressive, anti-inflammatory, and antimicrobial activities [11,12,13,14,15].

The existing studies have rarely identified lactones, which often have better pharmacological activities including anti-inflammatory activities [16,17,18]. Such constituents are mainly obtained through chemical synthesis or microbial biosynthesis, which involve many pathways and enzymes [19,20]. Previous studies on *A. crenata* Sims have focused on the root of *A. crenata* Sims, however, the non-medicinal parts of *A. crenata* Sims have been less studied. In this report, in order to expand the available resources and search for new bioactive constituents of *A. crenata* Sims, we described the isolation and structural elucidation of lactones and flavonoids. Moreover, the antibacterial and anti-inflammatory activities of two new lactones are discussed by the paper diffusion method and enzyme-linked immunosorbent (ELISA) assay.

## 2. Results and Discussion

### 2.1. Structure Elucidation

Compound **1** was separated as a light yellow amorphous solid. The molecular formula was inferred as C_26_H_28_O_11_ according to the HR-ESI-MS analysis of *m/z* 539.1522 [M + Na]^+^ (calculated value 539.1524 [M + Na]^+^), and the calculated unsaturation (Ω) = 13. The ^1^H-NMR spectrum of compound **1** (Table 1) showed two aromatic ring proton signals *δ*_H_ 7.49 (2H, dd, *J* = 1.9, 7.8 Hz, H-2‴, H-6‴); 7.26 (3H, *m*, H-3‴, H-4‴, H-5‴); *δ*_H_ 6.25 (1H, *d*, *J* = 2.9 Hz, H-3′); 6.19 (1H, *d*, *J* = 2.9 Hz, H-5′), a double bond signal *δ*_H_ 7.06 (1H, *d*, *J* = 12.6 Hz, H-7‴); 5.98 (1H, *d*, *J* = 12.6 Hz, H-8‴) identified as *cis* by the coupling constant, together with one anomeric proton at *δ*_H_ 4.44 (1H, *d*, *J* = 7.6 Hz, H-1″). ^13^C-NMR spectrum data (Table 1) revealed two benzene ring carbons and one pair of olefin carbons *δ*_C_ 103.4, 109.2, 120.3, 129.1, 130.0, 130.7, 133.0, 136.4, 138.5, 144.8, 151.5, 155.9, two ester carbonyl data *δ*_C_ 167.6, 180.5, and one sugar unit *δ*_C_ 64.5, 71.3, 75.4, 75.9, 77.9, 107.5, and the remaining four carbons were identified by DEPT 135° as three methylene *δ*_C_ 28.3, 29.5, 82.8 and one hypomethyl *δ*_C_ 36.5. Thus, the skeleton was identified as a five-membered lactone ring fragment by the key ^1^H-^1^H COSY signal (Figure 1). The connection of each fragment was determined by key HMBC signals (Figure 1) including H_2_-3 and H_1_-5 to C-2; H_2_-6 to C-4, C-1′, C-5′ and C-6′; H_1_-6″ to C-5″, C-4′ and C-9‴; H_1_-8‴ to C-9‴; H_1_-7‴ to C-2‴ and C-9‴. The spectroscopic data of compounds **1** (Appendix A) were available as Appendix A. The structures are shown in Figure 2. The structure of compound **1** was similar to that of Myrsinoside A [21], differing in the cinnamoyl and pentadactyl lactone ring portions.

Acid hydrolysis of **1** provided D-glucose. In addition, the coupling constant of H_2_-1″, *J* = 7.6 Hz indicated the configuration of the hydroxyl group at the anomeric carbon in sugar to be β. To further elucidate its absolute configuration, a combined CD spectrum with the electronic circular dichroism (ECD) spectrum of **1** recorded in MeOH showed a negative cotton effect at 208 nm, a positive cotton effect at 250 nm, a negative cotton effect at 267 to 296 nm, and a positive cotton effect at 297 to 325, which was broadly consistent with the calculated ECD data of the (5S) model (Figure 3). The calculation process could be found in the Appendix A. Thus, the structure of **1** was established as ((2R,3S,4S,5R,6S)-6-(2,4-dihydroxy-6-(((S)-5-oxotetrahydrofuran-2-yl) methyl) phenoxy)-3,4,5-trihydroxytetrahydro-2*H*-pyran-2-yl) methyl (*Z*)-3-phenylacrylate and was named Ardisicreolides A (Figure 1).

Compound **2** was obtained as a colorless amorphous solid. HR-ESI-MS analysis revealed the molecular formula to be C_26_H_28_O_11_ based on the [M + Na]^+^ signal at *m/z*: 539.1525 (calculated value of 539.1524 [M + Na]^+^) with the calculated unsaturation (Ω) = 13. The ^1^H-NMR (Table 1) spectrum of compound **2** exhibited two aromatic ring proton signals *δ*_H_ 7.61 (2H, *dd*, *J* = 1.9, 7.8 Hz, H-2‴, H-6‴); 7.42 (3H, *m*, H-3‴, H-4‴, H-5‴); *δ*_H_ 6.23 (1H, *d*, *J* = 2.9 Hz, H-3′); 6.18 (1H, *d*, *J* = 2.9 Hz, H-5′); a double bond signal *δ*_H_ 7.73 (1H, *d*, *J* = 16.0 Hz, H-7‴); 6.54 (1H, *d*, *J* = 16.0 Hz, H-8‴) identified as *trans* by the coupling constant, together with one anomeric proton *δ*_H_ 4.57 (1H, *d*, *J* = 7.6 Hz, H-1″). The ^13^C-NMR spectrum data (Table 1) showed two benzene ring carbons and one pair of olefin carbons *δ*_C_ 103.4, 109.2, 118.5, 129.4, 130.1, 131.6, 132.9, 135.7, 138.6, 146.8, 151.6, 155.9; two ester carbonyl data *δ*_C_ 168.3, 180.4; and one sugar unit data *δ*_C_ 64.7, 71.5, 75.4, 76.0, 77.9, 107.6; and the remaining four carbons were identified by DEPT135° as three methylene *δ*_C_ 28.3, 29.5, 82.8 and one hypomethyl *δ*_C_ 36.6. Thus, the skeleton was identified as a five-membered lactone ring fragment by the key ^1^H-^1^H COSY signal (Figure 1). Through a data comparison, compound **2** was similar to compound **1** and was determined to contain a cinnamoyl fragment, a sugar fragment, and an aromatic ring fragment, differing in the five-membered lactone ring portion. The connection mode of each segment was determined by key HMBC signals (Figure 1) including H_2_-3 and H_1_-5 to C-2; H_2_-6 to C-4, C-1′, C-5′ and C-6′; H_1_-6″ to C-5″, C-4″ and C-9‴; H_1_-8‴ to C-9‴; H_1_-7‴ to C-2‴, and C-9‴. The spectroscopic data of compounds **2** (Appendix A) were available as Appendix A. The structure is shown in Figure 2. The structure of compound **2** was similar to that of Myrsinoside A [21]. 

The conformation of the sugar was inferred to be β-D-glucose based on ^1^H-NMR and ^13^C-NMR by comparison with the known literature, and the acid hydrolysis of glucose confirmed that the sugar contained in compound **2** was D-glucose, and the relative conformation of the sugar was determined to be β due to the H_2_-1″ coupling constant of 7.6 Hz, so it was β-D-glucose. By comparing the 1D and 2D NMR spectra, the absolute configuration of compound **2** at the 5-position was the same as that of compound **1**, which was confirmed by CD spectroscopy, so the structure of compound **2** was characterized as ((2R,3S,4S,5R,6S)-6-(2,4-dihydroxy-6-(((S)-5-oxotetrahydrofuran-2-yl) methyl) phenoxy)-3,4,5-trihydroxytetrahydro-2*H*-pyran-2-yl) methyl (*E*)-3-phenylacrylate and named as Ardisicreolide B.

The known compounds (**3**–**6**) were identified on the basis of a detailed spectroscopic interpretation in comparison to the reported data in the references, to be Quercetin (**3**) [22], Myricetrin (**4**) [23], Quercitrin (**5**) [24], and Tamarixetin 3-O-rhamnoside (**6**) [25] (Figure 1). Their ^1^H-NMR (400 MHz) and ^13^C-NMR (100 MHz) data were in the Appendix A.

### 2.2. Effect of Compounds ***1**–**6*** on Antibacterial Activities

The inhibitory activities of compounds **1**–**6** were determined by measuring the diameter of the inhibition zone against six bacteria (*Escherichia coli*, *Bacillus subtilis*, *Staphylococcus aureus*, *Enterococcus faecalis*, *Pseudomonas aeruginosa*, and *Pseudomonas aeruginosa*) with 50 μg/mL ceftiofur sodium (Cef) as a positive control. The results showed that Ardisicreolide B had a better inhibition effect on *Bacillus subtilis* (Table 2).

### 2.3. Effects of Compounds ***1**–**2*** on RAW264.7 Cells by CCK-8 Method

To find anti-inflammatory compounds, the effect of Ardisicreolides A and B on the cell viability of LPS-stimulated RAW264.7 cells was evaluated with the cck-8 reagent. LPS activation of RAW246.7 cells could result in enhanced activity, and positive drug and compounds **1**–**6** were compared with the LPS-stimulated group alone. It was found that Ardisicreolide A had significant anti-inflammatory activity at ≥20 μM/mL, and Ardisicreolide B at ≥40 μM/mL showed significant anti-inflammatory activity (Table 3). Therefore, Ardisicreolides A and B were tentatively identified as compounds with potential anti-inflammatory activity (Figure 4), IC_50_ see Table 3.

### 2.4. Effects of Compounds ***1**–**2*** on NO Production

NO has a variety of regulatory effects on inflammation, and plays an increasingly important role in mediating inflammatory response. Therefore, Griess reagents were used to measure the effect of Ardisicreolides A and B on NO production in LPS-stimulated RAW264.7 cells, and dexamethasone was used as a positive control to evaluate the anti-inflammatory activity of the compounds. As a result (Appendix A), both Ardisreolides A and B showed inhibitory activity on the amount of NO release (Figure 5), however, the activity could not correlate significantly with the compound concentration.

### 2.5. Effects of Compounds ***1**–**2*** on Inflammatory Cytokines Production

IL-1β, IL-4, IL-10, and TNF-α are important inflammatory regulators produced in the process of inflammatory response. They can be produced and released in large amounts under the conditions of infection, injury, and immune response [26]. Therefore, they are commonly used as indicators to assess the anti-inflammatory effects of natural compounds [27,28]. In this paper, the effects of new compounds on the production of IL-1β, IL-4, IL-10, and TNF-α release by LPS-stimulated RAW264.7 cells were quantified with ELISA kits. As shown in Figure 6 (Appendix A), Ardisicreolides A and B had significant effects on IL-1β, IL-4, IL-10, and TNF-α at ≥5 μM/mL. This indicates that both Ardisicreolides A and B could effectively inhibit the inflammatory response of the LPS-stimulated RAW264.7 cells (Figure 6).

## 3. Materials and Methods

### 3.1. General Experimental Procedures

The 1D and 2D NMR spectra were recorded on a Bruker DPX 400 instrument (Bruker, Bremen, Germany) with tetramethylsilane as the internal standard and MeOH-d4 as the solvent. The HR-ESI-MS experiments were performed on a Waters Xevo G2-S QTOF (Waters Corporation, Milford, MA, USA). The semi-preparative HPLC procedure was performed on a Shimadzu LC-16P instrument with a RID-20A (Shimadzu Tokyo, Japan) and a reversed-phase C18 column (250 × 10 mm, 5 µm, Shim-pack GIST, Shimadzu Tokyo, Japan)). UV spectra were scanned with a SHIMADZU UV-2401PC spectrometer (Shimadzu Tokyo, Japan). Infrared spectra were performed on a VERTEX 70 spectrometer (Bruker, Bremen, Germany) using KBr particles. The rotational luminosity was measured on an Autopol VI instrument. Electron circular dichroism spectra were recorded on a LAAPD detector. Column chromatography was performed with silica gel (200–300 mesh, Qingdao Marine Chemical Ltd., Qingdao, China) and octadecyl silica gel (ODS) (50 µm, Merck, Darmstadt, Germany).

### 3.2. Plant Material

The leaves of *Ardisia crenata* Sims were collected from Guiyang in Guizhou Province (China), and identified by Professor Sheng-hua Wei from Guizhou University of Traditional Chinese Medicine. The voucher specimen (Accession number: 20200908) was deposited at the Guizhou University of Traditional Chinese Medicine.

### 3.3. Extraction and Isolation

The dried leaves (5 kg) of *Ardisia crenata* Sims were crushed, extracted with 70% ethanol at reflux for three times, the solvent was recovered under reduced pressure to obtain the crude extract, and the suspension was obtained by adding 10 L of distilled water. The crude extract was partitioned with petroleum ether, EtOAc, and n-BuOH successively to yield EtOAc (522.0 g) extracts. The soluble fraction of the EtOAc (220.8 g) was eluted by dichloromethane-methanol (0:1–1:0) on a silica gel column for a total of ten fractions (Fr. A–J). Fraction D (5.0 g) was chromatographed on an ODS column with MeOH-H_2_O (1:9 to 1:0) to afford sub-fractions D1–D9, Fr. D5 was eluted by MCI with MeOH-H_2_O (3:7 to 1:0) to afford sub-fractions D51–D58 and Fr. D55 was separated by semi-preparative HPLC (MeOH-H_2_O, 62:38; flow rate: 3 mL·min^−1^) to obtain compound **1** (5.9 mg t_R_ = 13.7 min), compound **2** (3.6 mg t_R_ = 19.5 min), and compound **5** (9.0 mg t_R_ = 21.4 min). Fraction E (9.6 g) was chromatographed on an ODS column with MeOH-H_2_O (1:9 to 1:0) to afford sub-fractions E1–E9, Fr. E4 was separated by semi-preparative HPLC (MeOH-H_2_O, 55:45, flow rate: 3 mL·min^−1^) to obtain compound **6** (6.8 mg t_R_ =15.0 min). Fraction I (17.6 g) was chromatographed on an ODS column with MeOH-H_2_O (1:9 to 1:0) to afford sub-fractions I1–I9, Fr. I4 was separated by semi-preparative HPLC (MeOH-H_2_O, 45:55; flow rate: 3 mL·min^−1^) to obtain compound **3** (7.2 mg t_R_ = 20.0 min) and compound **4** (6.8 mg t_R_ = 34.2 min).

### 3.4. Characterization of Compounds ***1**–**2***

Ardisicreolide A, Yellowish amorphous solid; ${\left[\alpha \right]}_{D}^{24}$ + 14.23 (c 0.05, MeOH); UV (MeOH) λ (logε) 203 (4.29) nm; HR-ESI-MS *m*/*z* 539.1522 [M + Na]^+^ (Calcd for C_26_H_28_O_11_, 539.1524); ^1^H and ^13^C NMR data (CD_3_OD), see Table 1.

Ardisicreolide B, Colorless amorphous solid; ${\left[\alpha \right]}_{D}^{24}$ − 22.65 (c 0.08, MeOH); UV (MeOH) λ (logε) 203 (4.45) nm; HR-ESI-MS *m*/*z* 539.1525 [M + Na]^+^ (Calcd for C_26_H_28_O_11_, 539.1524); ^1^H and ^13^C NMR data (CD_3_OD) see Table 1. 

### 3.5. Antibacterial Activity Screening

The blank drug sensitive test paper was dipped to 50 μg/mL of ceftiofur sodium solution and the compound solution of each concentration, respectively. A total of 100 μL of the test solutions (*Escherichia coli*, *P. aeruginosa*, *Enterococcus faecalis*, *Proteusvulgaris*, *Staphylococcus aureus*, and *Bacillus subtilis*) were absorbed respectively, and LB medium was spread evenly using a sterile applicator stick, and took the drug-sensitive test paper with sterile forceps and spread it evenly on the center of the surface of the medium in the divided area correspondingly. Then, the medium was put in a constant temperature incubator until the drug solution adsorbed into the medium, and was incubated at 37 °C for 8 h, and measured the diameter of the inhibition zone by repeating it three times and recording the data.

### 3.6. Cell Culture

The RAW264.7 cell lines were obtained from the Chinese Academy of Sciences (Shanghai, China) cell bank and cultured in DMEM supplemented with 10% (*v*/*v*) heat-inactivated FBS at 37 °C in a fully humidified incubator containing 5% CO_2_. The cells were passaged when they grew to 80–90% confluence.

### 3.7. Primary Screening for Anti-Inflammatory Compounds

The RAW264.7 cells were inoculated in complete medium containing double antibodies in 10% FBS at 37 °C with 5% CO_2_ in a constant temperature incubator, and 96-well plates were inoculated with cell suspensions (100 uL/well) at a density of 10^5^/mL at 37 °C with 5% CO_2_ in a constant temperature incubator. Then, cells were pretreated with different isolated compounds (5–160 μM) or dexamethasone (40 μg/mL) for 1 h and then stimulated with LPS (1 μg/mL). The normal and model cells were stimulated with or without 1 μg/mL LPS for 24 h. All samples were tested in quadruplicate according to the CCK-8 kit after incubation.

### 3.8. Determination of Inflammatory Cytokines

The RAW264.7 cells were plated in 24-well plates and incubated at 37 °C and 5% CO_2_ for 24 h. Then, cells were pretreated with different isolated compounds (5, 20, 80 μM) or dexamethasone (40 μg/mL) for 1 h and then stimulated with LPS (1 μg/mL). The normal and model cells were stimulated with or without 1 μg/mL LPS for 24 h. The supernatants of the RAW264.7 cells after drug administration and culture were used to determine the NO release according to the Griess method, and the contents of IL-1β, IL-4, IL-10, and TNF-α in the cells were determined according to enzyme-linked immunosorbent assay kits (ELISA) following the manufacturer’s instructions [29], and all samples were taken in triplicate.

## 4. Conclusions

In summary, a total of six compounds were isolated and identified from the EtOAc fraction of the 70% ethanol extract of the leaves of *Ardisia crenata* Sims including two new lactone structures Ardisicreolides A and B, which were analyzed and identified mainly by MS, NMR, and IR. In previous studies, *A. crenata* Sims components focused on triterpene saponins, isocoumarins, and benzoquinones, however, lactones were rarely identified. In this paper, we evaluated the antibacterial and anti-inflammatory activities of two new lactones isolated and identified, in which Ardisicreolide B showed good inhibition of *Bacillus subtilis* at ≥50 μg/mL (IZD = 13.2 ± 1.01 mm), and both Ardisicreolides A and B exhibited significant anti-inflammatory activity against inflammatory factors of NO, IL-1β, IL-4, IL-10 and TNF-α release with varying degrees. Therefore, the lactone components in the leaves of *A. crenata* Sims might be natural effective drugs for anti-inflammatory drug development, and the non-traditional medicinal parts of *A. crenata* Sims could be identified as a source of natural anti-inflammatory molecules, which is of great significance for the rational development and application of *A. crenata* Sims.

## Figures and Tables

**Figure 1 molecules-27-04903-f001:**
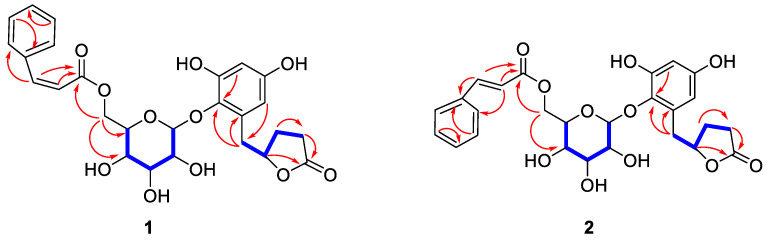
The key HMBC (H→C) and ^1^H-^1^H COSY correlations of compounds **1**–**2**.

**Figure 2 molecules-27-04903-f002:**
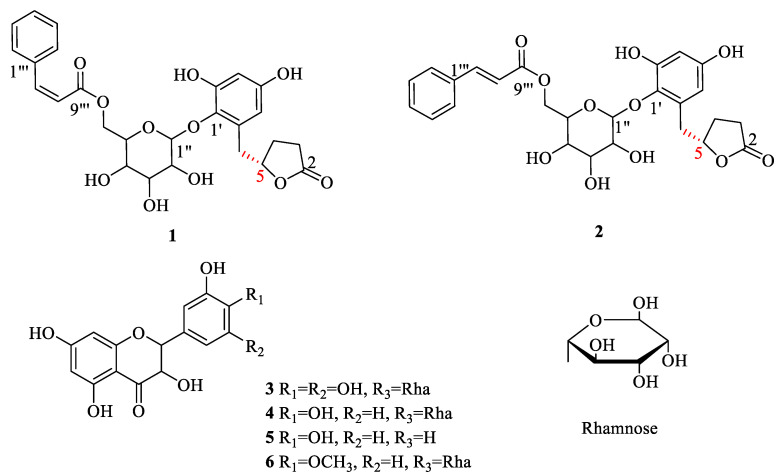
The structures of compounds **1**–**6** from the leaves of *A. crenata* Sims.

**Figure 3 molecules-27-04903-f003:**
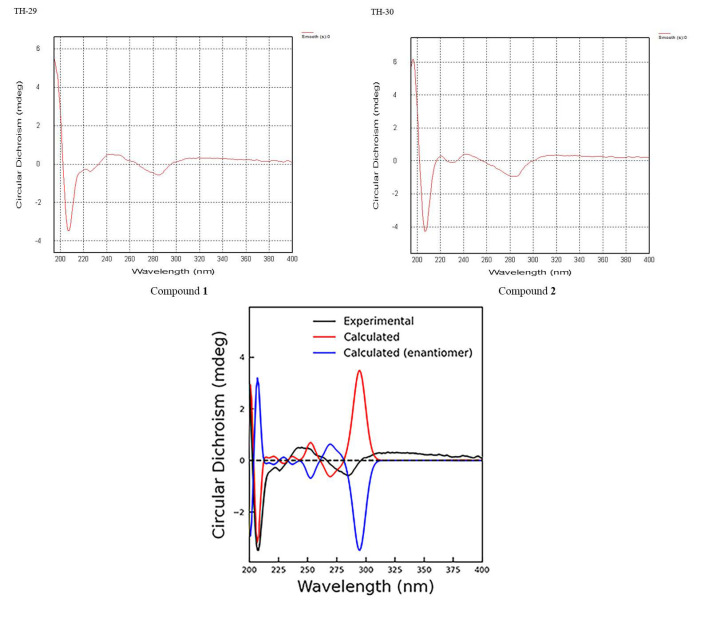
The CD and ECD spectra of compounds **1**–**2**.

**Figure 4 molecules-27-04903-f004:**
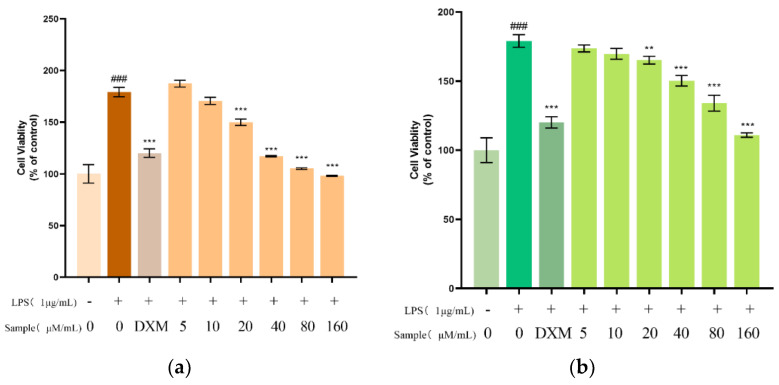
The effects of Ardisicreolides A (**a**) and B (**b**) on the cell viability of LPS-treated RAW264.7 cells. The data were expressed as the mean ± SD (*n* = 3). ** *p* < 0.01, *** *p* < 0.001 versus the control cells that were treated with LPS. ^###^ *p* < 0.001 versus the control group.

**Figure 5 molecules-27-04903-f005:**
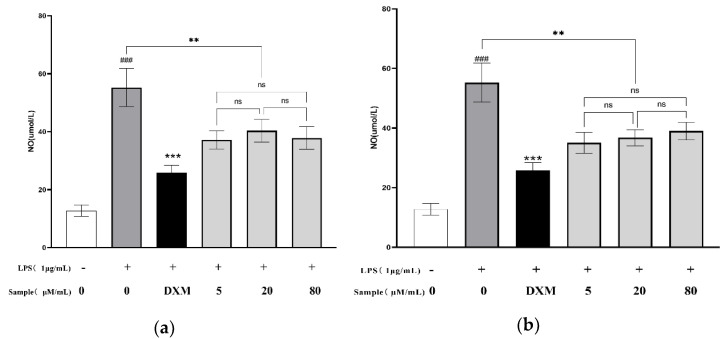
Ardisicreolides A (**a**) and B (**b**) decreased NO production. Dexamethasone with 1 μg/mL was used as the positive control. The data were expressed as the mean ± SD (*n* = 3). ns > 0.05, ** *p* < 0.01, *** *p* < 0.001 versus the model cells that were treated by LPS. ^###^ *p* < 0.001 versus the control.

**Figure 6 molecules-27-04903-f006:**
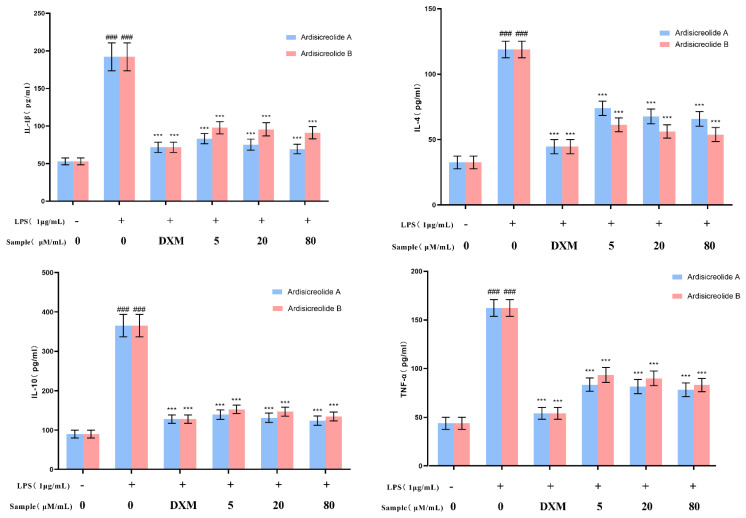
Ardisicreolides A and B decreased the inflammatory cytokine production. Dexamethasone with 40 μg/mL was used as the positive control. The data were expressed as the mean ± SD (*n* = 3). *** *p* < 0.001 compared to the model cells that were treated by LPS. ^###^
*p* < 0.001 versus the control.

**Table 1 molecules-27-04903-t001:** The ^1^H and ^13^C-NMR data of compounds **1**–**2** in CD_3_OD (*δ* in ppm).

Position	Ardisicreolide A			Ardisicreolide B		
	*δ_C_*	*δ* _H_	(*J* in Hz)	*δ_C_*	*δ* _H_	(*J* in Hz)
2	180.5	-	-	180.4	-	-
3	29.5	2.35	m	29.5	2.45	dd (7.5, 8.9)
4	28.3	1.90, 2.19	m, m	28.3	1.96, 2.23	dtd (7.0, 8.9, 12.5), m
5	82.8	4.76	m	82.8	4.84	*M*
6	36.5	2.85, 3.15	dd (4.7, 14.1), dd (8.0, 14.1)	36.6	2.87, 3.25	dd (5.2, 14.0), dd (7.6, 14.0)
1′	138.5	-	-	138.6	-	-
2′	151.5	-	-	151.6	-	-
3′	103.4	6.25	d (2.9)	103.4	6.23	d (2.9)
4′	155.9	-	-	155.9	-	-
5′	109.2	6.19	d (2.9)	109.2	6.18	d (2.9)
6′	133.0	-	-	132.9	-	-
1″	107.5	4.44	d (7.6)	107.6	4.57	d (7.6)
2″	75.4	3.28~3.46	m (4H)	75.4	3.37~3.62	m (4H)
3″	77.9			77.9		
4″	71.3			71.5		
5″	75.9			76.0		
6″	64.5	4.24, 4.50	dd (6.5, 12.0), dd (2.0,12.0)	64.7	4.34, 4.59	dd (6.5, 12.0), dd (2.0,12.0)
1‴	136.4	-	-	135.7	-	-
2‴ 6‴	130.7	7.49	dd (1.9, 7.8, 2H)	130.1	7.61	dd (1.9, 7.8, 2H)
3‴ 5‴	129.1	7.26	m (3H)	129.4	7.42	m (3H)
4‴	130.0			131.6		
7‴	144.8	7.06	d (12.6)	146.8	7.73	d (16.0)
8‴	120.3	5.98	d (12.6)	118.5	6.54	d (16.0)
9‴	167.6	-	-	168.3	-	-

**Table 2 molecules-27-04903-t002:** The antibacterial activities of Ardisicreolide B. (mean ± SD, *n* = 3) (d, mm).

C(μg/mL)	Ardisicreolide B			Cef ^a^
	25	50	100	
*Escherichia coli*	-	-	-	16.63 ± 0.99
*P. aeruginosa*	-	-	-	19.30 ± 1.42
*Bacillus subtilis*	11.33 ± 1.01	13.2 ± 1.01	17.47 ± 1.53	29.37 ± 1.01
*Enterococcus* *Faecalis*	-	-	-	19.40 ± 1.01
*Proteus vulgaris*	-	-	-	18.47 ± 0.78
*Staphylococcus aureus*	-	-	-	28.40 ± 1.35

^a^ Positive control.

**Table 3 molecules-27-04903-t003:** The IC_50_ values of Ardisicreolides A and B as inhibitors of LPS-treated RAW264.7 cells.

Compound	Mean ± SD (μM/mL)
Ardisicreolide A	24.46 ± 1.57
Ardisicreolide B	55.85 ± 4.28

## Data Availability

Not applicable.

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
