# Peer review of "Two New Phenolic Glycosides with Lactone Structural Units from Leaves of *Ardisia crenata* Sims with Antibacterial and Anti-Inflammatory Activities"

_molecules, 2022, doi:10.3390/molecules27154903_

Round 1

Reviewer 1 Report

Lactone is a very common term. Actually the compounds are simple phenol glycosides including a lactone unit. Therefore, I suggest to change the title using more explanatory title.

Compound 1 is the cis-derivative of the compound 2 and it can be an artefact occurred during the extraction and isolation process. It will be better to discuss firstly trans-form and a brief discussion will be sufficient indicating the cis-form of the cinnamoyl moiety.

Abstract: line 7: P. Aeuroginosa should be corrected as Pseudomonas aeuroginosa as other microorganisms.

lines 54 and 83. ... two aromatic ring protons  or ..... protons arising from the two aromatic rings... (?).

line 56: .....a double bond signal.....   or ...two olefinic protons as an AX system... 

line 59. ... 14 aromatic region carbon data...  This sentence can be formulated in a better manner.

lines 72-73. "th" and "e" should be combined.

line 76. 1 should be written as bold.

Reviewer 2 Report

Authors have done a very interesting study about new lactones from leaves of Ardisia crenata Sims.  Honestly, I have done many similar experiments and read many similar studies but I am  impressed about obtained results by the authors.  I commend the authors for this study.

Just only one thing missing in this research. It is the use of different model to describe anti inflammatory action of this two lactones e.g Anti-AChE or Anti-BuChE models. If is not possible to done it, it will be valuable if obtained results will be commentet in existing data.

Reviewer 3 Report

Dear authors,

please find my recommendation enclosed.

Best

Round 2

Reviewer 1 Report

Thank you very much for your kind response.

Author Response

Dear editors,

Thanks for your letter. Thank you for your help and suggestions in the process of revising the article. 

May you have a happy life and smooth work. 

With best regards,

Huihui Tao

Email: thlc922@163.com

Author Response

Dear reviewer,

Thanks for your letter. Thank you for your help and suggestions in the process of revising the article. Your suggestions  have been revised in the manuscript and highlighted in blue font in this time.

May you have a happy life and smooth work. 

With best regards,

Huihui Tao

Email: thlc922@163.com

This manuscript is a resubmission of an earlier submission. The following is a list of the peer review reports and author responses from that submission.

Round 1

Reviewer 1 Report

This manuscript entitled Two new lactones from leaves of Ardisia crenata Sims with antibacterial and anti-inflammatory activities by Zhou et al. described, for the first time, the isolation and characterisation of the two new lactones along with four known compounds from this plant. 

The structures of these lactones were determined by spectroscopic analysis using HR-MS, 1D and 2D NMR, ECD. Biolocigal activity assays including antibacterial and anti-inflammatory were performed and the result showed that compounds 1 and 2 were active in the NO and cytokines production assays while all the other isolated compounds have no biologacal activity. Despite the authors have done an excellent job for the isolation and structure elucidation of two new lactones, the novelty and the scope of study of the manuscript as a whole is not very significant. The structures of the isolated compounds are similar to the known compound myrsinoside A.

Besides, the authors did not specify the sterochemistry of the sugar  in compound 1 and 2 in the Figure 1 and the numbering carbons were missed out in both structures which will make it more challenging for the readers to read through the manuscript. An additional question would be 'how the authors be certain that the B-lactones moiety in compound 1 and 2 are connected to the 1-position of the sugar molecules where the COSY did not provide this data?' The biosynthesis of these compounds were also not discussed as the B-lactones in both structures are somewhat unusual stuctures for natural product. 

Reviewer 2 Report

The manuscript entitled “Two new lactones from leaves of Ardisia crenata Sims with antibacterial and anti-inflammatory activities” describes the discovery of two new lactones from leaves of Ardisia crenata Sims. The structures were elucidated by the analyses of 1D and 2D-NMR data and comparison with spectral data from published literature. Compounds 1 and 2 were found to inhibit the production of NO and reduce the release of TNF-α, IL-1β, IL-4 and IL-10 in LPS-induced RAW 264.7 macrophage cells. Although the structure elucidation seems logical and the activities look promising, the paper itself was poorly written. Looks like the authors did take the manuscript preparation seriously. Thus, I suggest that this manuscript should be rejected unless authors provide a well-written version.

Before I dig into the details of the results presented in this manuscript, typos should be addressed by the authors. Actually, there are so many, so I can only list a few. For now, the manuscript is unreadable.

Abstract:

“….Ethyl acetate fractions…” usually say ethyl acetate “portion”.

“….cps 3-6 were isolated from this plant…”??? I assume authors want to say these compounds were identified from Ardisia crenata Sims for the first time. Please rephrase it.

”…good anti…” we usually don’t say good in a paper. Please use “moderate” or something else.

“…markedly…” what’s markedly?

Please rephrase “The structures of 1-6 were elucidated by analysis of 1D and 2D-NMR data and comparison with spectral data from published literature.” Maybe should just focus on the methods applied for the structural elucidation of the new compounds.

Please revise “In this study, all compounds were evaluated antibacterial activity in vitro by determination of the inhibition zone of Escherichia coli, P.Aeruginosa, Enterococcus faecalis, Proteus vulgaris, Staphylococcus aureus and Bacillus subtilis.” into “All the isolated compounds were evaluated for in vitro anti-microbial effect against Escherichia coli, P.Aeruginosa, Enterococcus faecalis, Proteus vulgaris, Staphylococcus aureus and Bacillus subtilis.”

Line 50: m/z into m/z

58: 2 ester…-----two ester

58: carbonyl data???? Carbonyl resonances at…right?

67: “the conformation….” Please rephrase this sentence.

Please go through the whole manuscript and revise accordingly.